# PrivilegedDreamer: Explicit Imagination of Privileged Information for Adaptation in Uncertain Environments

## Abstract

Numerous real-world control problems involve dynamics and objectives affected by unobservable hidden parameters, ranging from autonomous driving to robotic manipulation. To represent these kinds of domains, we use Hidden-parameter Markov Decision Processes (HIP-MDPs), which model sequential decision problems where hidden variables parameterize transition and reward functions. Existing approaches, such as domain randomization, domain adaptation, and meta-learning, simply treat the effect of hidden parameters as additional variance and often struggle to effectively handle HIP-MDP problems, especially when rewards are parameterized by hidden variables. To address this, we introduce Privileged-Dreamer, a model-based reinforcement learning framework that extends Dreamer, a powerful world-modeling approach, by incorporating an explicit parameter estimation module. We introduce a novel dual recurrent architecture that explicitly estimates hidden parameters from limited historical data and enables us to condition the model, actor, and critic networks on these estimated parameters. Our empirical analysis on five diverse HIP-MDP tasks demonstrates that it outperforms state-of-the-art model-based, model-free, and domain adaptation learning algorithms. Furthermore, we also conduct ablation studies to justify our design decisions.

## 1 Introduction

The Markov Decision Process (MDP) has been a powerful mathematical framework for modeling a spectrum of sequential decision scenarios, from computer games to intricate autonomous driving systems; however, they often assume fixed transition or reward functions. In many real-world domains, there exists a family of related problems characterized by the presence of hidden or uncertain parameters that play a significant role in their dynamics or reward functions, which is referred to as a hidden-parameter MDP (HIP-MDP) (Doshi-Velez & Konidaris, 2016). For instance, autonomous driving must deal with diverse vehicles with distinctive dynamic attributes and properties for better driving experience, while the agricultural industry sorts produce that varies in weight. Consequently, research endeavors have explored diverse algorithmic approaches, including domain randomization (Tobin et al., 2017), domain adaptation (Peng et al., 2020), and meta-learning (Wang et al.), to address these challenges effectively.

We approach these HIP-MDP problems using model-based reinforcement learning because a world model holds significant promise in efficiently capturing these dynamic behaviors characterized by hidden parameters, ultimately resulting in improved policy learning. Particularly, we establish our framework based on Dreamer (Hafner et al., 2019), which has been effective in solving multiple classes of problems, including DM control suite (Tassa et al., 2018), Atari (Hafner et al., 2020), and robotic control (Wu et al., 2022). Our initial hypothesis was that the Dreamer framework may be able to capture parameterized dynamics accurately by conditioning the model on latent variables, leading to better performance at the end of learning. However, Dreamer is designed to predict action-conditioned dynamics in the observation space and does not consider the effect of hidden parameters.

This paper presents PrivilegedDreamer to solve HIP-MDPs via explicit prediction of hidden parameters. Our key intuition is that a recurrent state space model (RSSM) of model-based RL must be explicitly conditioned on hidden parameters to capture the subtle changes in dynamics or rewards. However, a HIP-MDP assumes that hidden variables are not available to agents. Therefore, we introduce an explicit module to estimate hidden parameters from a history of state variables via a Long short-term memory (LSTM) network, which can be effectively trained by minimizing an additional reconstruction loss. This dual recurrent architecture allows accurate estimation of hidden parameters from a short amount of history. The estimated hidden parameters are also fed into the transition model, actor, and critic networks to encourage their adaptive behaviors conditioned on hidden parameters.

We evaluate our method in five HIP-MDP environments, where two of them have parameter-conditioned reward functions. We compare our method against several state-of-the-art baselines, including model-based (DreamerV2 (Hafner et al., 2020)), model-free (Soft Actor Critic (Haarnoja et al., 2018) and Proximal Policy Optimization (Schulman et al., 2017)), and domain adaptation (Rapid Motor Adaptation (Kumar et al., 2021)) algorithms. Our PrivilegedDreamer achieves 41% higher average rewards over five tasks, particularly on HIP-MDPs with parameterized reward functions. We further analyze the behaviors of the learned policies to investigate how rapid estimation of hidden parameters affects the final performance and also to justify the design decisions of the framework. Finally, we outline a few interesting future research directions.

## 2 RELATED WORK

**World Models**   Model-based RL improves sample efficiency over model-free RL by learning an approximate model for the transition dynamics of the environment, allowing for policy training without interacting with the environment itself. However, obtaining accurate world models is not straightforward because the learned model can easily accumulate errors exponentially over time. To alleviate this issue, Chua et al. (2018) designs ensembles of stochastic dynamics models to attempt to incorporate uncertainty. The Dreamer architecture (Hafner et al., 2019; 2020; 2023) models the environment using the recurrent state space machine, which also includes the recurrent GRU network (Cho et al., 2014) and the VAE (Kingma & Welling, 2013), via reconstructing the input from a latent space. With this generative world model, the policy is trained with imagined trajectories in this learned latent space. Robine et al. (2023) and Micheli et al. (2022) leverage the Transformer architecture (Vaswani et al., 2017) to autoregressively model the world dynamics and similarly train the policy in latent imagination. Our work is built on top of the Dreamer architecture, but the idea of explicit modeling of hidden parameters has the potential to be combined with other architectures.

**Randomized Approaches without Explicit Modeling**   One of the most popular approaches to deal with uncertain or parameterized dynamics is domain randomization (DR), which aims to improve the robustness of the policy by exposing the agent to randomized environments. It has been effective in many applications, including manipulation (Peng et al., 2018; Tobin et al., 2017; Zhang et al., 2016; James et al., 2017), (Tobin et al., 2017), locomotion (Peng et al., 2020; Tan et al., 2018), autonomous driving (Tremblay et al., 2018), and indoor drone flying (Sadeghi & Levine). Domain randomization has also shown great success in deploying trained policies on actual robots, as in Tan et al. (2018), which used it for sim-to-real transfer for a quadrupedal robot, and Peng et al. (2018), which used it to improve performance for a robotic manipulator. While DR works very well in many situations, it tends to lead to an overly conservative policy that is suboptimal for challenging problems with a wide range of transition or reward functions.

**Domain Adaptation**   Another common strategy for dealing with variable environments is to incorporate the hidden environmental parameters into the policy for adaptation. This privileged information of the hidden parameters can be exploited during training, but at test time, system identification must occur online. For model-free RL, researchers typically train a universal policy conditioned on hidden parameters and estimate them at testing time by identifying directly from a history of observations (Yu et al., 2017; Kumar et al., 2021; Nahrendra et al., 2023). Another option is to improve state estimation while training in diverse environments, which similarly allows for adaptation without needing to perform explicit system identification (Ji et al., 2022). For model-based RL, the problem of handling variable physics conditions is handled in multiple ways. A few research groups

Nagabandi et al. (2018); Sæmundsson et al. (2018) propose using meta-learning to rapidly adapt to environmental changes online. Wang & van Hoof (2021) uses a graph-based meta RL technique to handle changing dynamics. Ball et al. (2021) used data augmentation in offline RL to get zero-shot dynamics generalization. The most applicable methods for our work are the problems that use a learned encoder to estimate a context vector that attempts to capture the environmental information and is used to condition the policy and for forward prediction, as in Wang et al. (2022); Lee et al. (2020); Huang et al. (2021); Seo et al. (2020).

## 3 PRIVILEGEDDREAMER: ADAPTATION VIA EXPLICIT IMAGINATION

### 3.1 BACKGROUND

**Hidden-parameter MDP** A Markov decision process (MDP) formalizes a sequential decision problem, which is defined as a tuple $(S, A, T, R, p_0)$, where $S$ is the state space, $A$ is the action space, $T$ is the transition function, $R$ is the reward function, and $p_0$ is the initial state distribution. For our work, we consider the hidden-parameter MDP (HIP-MDP), which generalizes the MDP by conditioning the transition function $T$ and/or the reward function $R$ on an additional hidden latent variable $\omega$ sampled from a distribution $p_\omega$ (Doshi-Velez & Konidaris, 2016). Without losing generality, $\omega$ can be a scalar or a vector. In the setting of continuous control, which is the primary focus of this work, this latent variable represents physical quantities, such as mass or friction, that govern the dynamics but are not observable in the state space.

**Dreamer** For our model, we build upon the DreamerV2 model of Hafner et al. (2020). DreamerV2 uses a recurrent state space model (RSSM) to model dynamics and rewards. This RSSM takes as input the state $x_t$ and the action $a_t$ to compute a deterministic recurrent state $h_t = f_\phi(h_{t-1}, z_{t-1}, a_{t-1})$ using a GRU $f_\phi$ and a sampled stochastic state $z_t \sim q_\phi(z_t|h_t, x_t)$ using an encoder $q_\phi$. The combination of these deterministic and stochastic states is used as a representation to reconstruct the state $\hat{x}_t \sim p_\phi(\hat{x}_t|h_t, z_t)$, and to also predict the reward $\hat{r}_t \sim p_\phi(\hat{r}_t|h_t, z_t)$ and the discount factor $\hat{\gamma}_t \sim p_\phi(\hat{\gamma}_t|h_t, z_t)$. The final component of the RSSM is the transition predictor $\hat{z}_t \sim p_\phi(\hat{z}_t|h_t)$. This computes the stochastic state $z_t$ using only the deterministic state $h_t$, which is necessary for training in imagination where the state $x_t$ is not available.

For policy learning, Dreamer adopts an actor-critic network, which is trained via imagined rollouts. For each imagination step $t$, the latent variable $\hat{z}_t$ is predicted using only the world model, the action is sampled from the stochastic actor: $a_t \sim \pi_\theta(a_t|\hat{z}_t)$, and the value function is estimated as: $v_\psi \approx \mathbb{E}_{p_\phi, p_\theta}[\sum \gamma^{\tau-t}\hat{r}_\tau]$, where $\hat{r}_t$ is computed from the reward predictor above. The actor is trained to maximize predicted discounted rewards over a fixed time horizon $H$. The critic aims to accurately predict the value from a given latent state. The actor and critic losses are:

$$\text{Actor loss: } L = \mathbb{E}_{p_\phi, p_\theta}\big[\sum_{t=1}^{H-1} -\ln \pi_\theta(\hat{a}_t|\hat{z}_t)sg(V_t^\lambda - v_\psi(\hat{z}_t)) - \eta H[a_t|\hat{z}_t]\big]$$

$$\text{Critic loss: } L = \mathbb{E}_{p_\phi, p_\theta}\big[\sum_{t=1}^{H-1} \frac{1}{2}(v_\psi(\hat{z}_t) - sg(V_t^\lambda))^2\big]$$

### 3.2 ALGORITHM

While the original DreamerV2 layout works effectively for many tasks, in the HIP-MDP domain, it falters, especially in the case where the reward explicitly depends on the hidden latent variable. Even though the RSSM has memory to determine the underlying dynamics, prior works such as Seo et al. (2020) have shown that this hidden state information is poorly captured implicitly and must be explicitly learned.

**Explicit Imagination via LSTM** To help remedy this, we incorporate an additional independent module for estimating the privileged information from the available state information. This dual recurrent architecture allows us to effectively estimate the important hidden parameters in the first layer and model other variables conditioned on this estimation in the second layer. Our estimation module $\tilde{\omega}_t \sim \eta_\phi(\tilde{\omega}_t|x_t, a_{t-1})$ takes the state $x_t$ and previous action $a_{t-1}$ as inputs and predicts the

intermediate hidden parameter $\tilde{\omega}_t$. It is still parameterized by $\phi$ because we treat it as part of the world model. The estimation module is comprised of an LSTM (Hochreiter & Schmidhuber) followed by MLP layers to reshape the output to that of the privileged data. We use an LSTM because its recurrent architecture is more suitable to model subtle and non-linear relationships between state and hidden variables over time. However, the choice of the architecture was not significant to the performance. In our experience, LSTM and GRU demonstrated similar performance.

Note that we use $\tilde{\omega}_t$ to make the recurrent world model conditioned on the estimated hidden variable. For the actor and critic, we feed the value from the prediction head, $\hat{\omega}_t$ which will be described in the next paragraph.

**Additional Prediction Head** We also added an additional prediction head $p_\phi(\hat{\omega}_t|h_t, z_t)$, which is similar to the reward or state prediction heads. While the previous LSTM estimation $\eta$ predicts the intermediate parameter $\tilde{\omega}_t$ to make the model conditioned on the hidden parameter, this additional prediction head offers two major improvements: 1) encouraging the RSSM state variables $h_t$ and $z_t$ to contain enough information about the hidden parameter and 2) improving the prediction accuracy.

**Hidden Variable Loss** We design an additional loss to train the estimation module, which is similar to the other losses of the DreamerV2 architecture. We do not use the discount predictor from the original DreamerV2 architecture as all of our tests are done in environments with no early termination. We group the other Dreamer losses all under $L_{Dreamer}$ to highlight our differences. This makes the total loss for the world model:

$$L(\phi) = L_{Dreamer} + \mathbb{E}_{q_\phi(z_{1:T}|a_{1:T}, x_{1:T}, \omega_{1:T})}[\sum_{t=1}^{T} -\ln \eta_\phi(\tilde{\omega}_t|x_t, a_{t-1}) - \ln p_\phi(\hat{\omega}_t|h_t, z_t)].$$

where the first loss is to compute an intermediate estimate $\tilde{\omega}$ for the hidden parameter $\omega$ using the environment states $x$ and actions $a$ and the second term is the world model reconstruction loss for $\hat{\omega}$ based on the RSSM latent variables $h$ and $z$.

It is important to highlight that relying solely on this hidden parameter loss term is not sufficient. Theoretically, it seems like the loss encourages the recurrent state variables $h_t$ and $z_t$ to encapsulate all relevant information and increase all the model, actor, and critic networks' awareness of hidden parameters. However, in practice, this privileged information remains somewhat indirect to those networks. Consequently, this indirect access hinders their ability to capture subtle changes and results in suboptimal performance.

**Hidden parameter conditioned Networks (ConditionedNet)** Once we obtain the estimated hidden parameter $\omega_t$, we feed this information to the networks. This idea of explicit connection has been suggested in different works in reinforcement learning, such as rapid motor adaptation (RMA) (Kumar et al., 2021) or meta strategy optimization (MSO) (Yu et al., 2020). Similarly, we augment the inputs of the representation model $z_t \sim q_\phi(z_t|h_t, x_t, \tilde{\omega}_t)$, the critic network $v_\psi$, and the actor network $\pi_\theta$ to encourage them to incorporate the estimated $\tilde{\omega}_t$ and $\hat{\omega}_t$.

**Additional Proprioceptive State as Inputs** In our experience, it is beneficial to provide the estimated state information as additional inputs to the actor and critic networks. We hypothesize that this may be because the most recent state information $x_t$ is highly relevant for our continuous control tasks. On the other hand, the RSSM states $h_t$ and $z_t$ are indirect and more suitable for establishing long-term plans.

**Summary** On top of DreamerV2, Our PrivilegedDreamer includes the following components:

$$\text{Recurrent hidden parameter predictor: } \tilde{\omega}_t \sim \eta_\phi(\tilde{\omega}_t|h_t, z_t)$$
$$\text{HIP-conditioned representation model: } z_t \sim q_\phi(z_t|h_t, x_t, \tilde{\omega}_t)$$
$$\text{HIP prediction head: } \hat{\omega}_t \sim p_\phi(\hat{\omega}_t|h_t, z_t)$$
$$\text{HIP-conditioned critic: } v_t \sim v_\psi(v_t|h_t, z_t, x_t, \hat{\omega}_t)$$
$$\text{HIP-conditioned actor: } \hat{a}_t \sim \pi_\theta(a_t|h_t, z_t, x_t, \hat{\omega}_t)$$

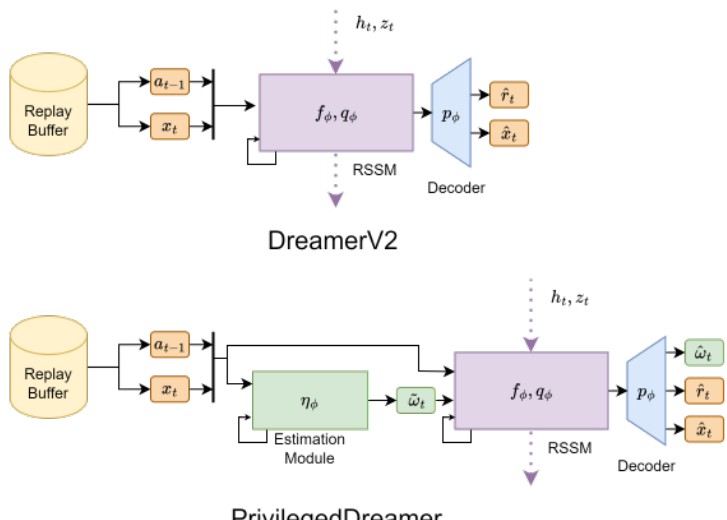

Figure 1: Architecture of the PrivilegedDreamer. Compared to the default DreamerV2 model (**top**), our architecture (**bottom**) adopts an explicit parameter estimation model $\eta$ to predict the hidden parameters $\omega_t$ from a history of states. Then, the estimated parameters $\tilde{\omega}_t$ are fed into the model to establish the explicit dependency.

We omit the unchanged components from DreamerV2, such as input and reward predictors, for brevity. A schematic of the model architecture used for training the world model itself can be seen in Figure 1. This setup trains the encoder network, decoder network, and the latent feature components $z$ and $h$. The estimation module $\eta$ that initially estimates the value of $\tilde{\omega}_t$ is also trained here.

For training the policy network in imagination, we use the structure in Figure 2. When training the policy, we start with a seed state sampled from the replay buffer and then proceed in imagination only, as in the original DreamerV2. Via this setup, the actor and critic networks are trained to maximize the estimated discounted sum of rewards in imagination using a fixed world model. However, the key difference is that both the actor and critic networks take the estimated parameter $\hat{\omega}_t$ from the prediction head as an additional input, as well as the reconstructed state $\hat{x}_t$. Because the entire

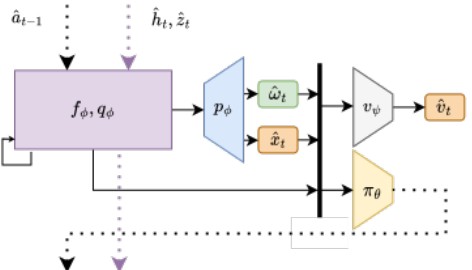

Figure 2: Policy network training architecture.

model can learn the parameter estimation much faster than the world model, this new connection works almost the same as providing the ground-truth hidden parameter for the majority of the learning time. We will discuss this behavior in the discussion section.

## 4 EXPERIMENTS

We evaluate PrivilegedDreamer on several HIP-MDP problems to answer the following research questions:

1. Can our PrivilegedDreamer solve HIP-MDP problems more effectively than the baseline RL and domain adaptation algorithms?

2. Can the estimation network accurately find ground-truth hidden parameters?

3. What are the impacts of the HIP reconstruction loss and hip-conditioned policy?

| Task | Physics Randomization Target | Range | Reward |
|---|---|---|---|
| Walker Run | Contact Friction | [0.05 - 4.5] | Fixed |
| Pendulum Swingup | Mass Scaling Factor of Pendulum | [0.1 - 2.0] | Fixed |
| Throwing | Mass Scaling Factor of Ball | [0.2 - 1.0] | Fixed |
| Sorting | Mass Scaling Factor of Arm | [0.2 - 1.0] | Parameterized |
| Pointmass | X/Y Motor Scaling Factor | X [1 - 2] Y [1 - 2] | Parameterized |

Table 1: Parameter randomization applied for each task.

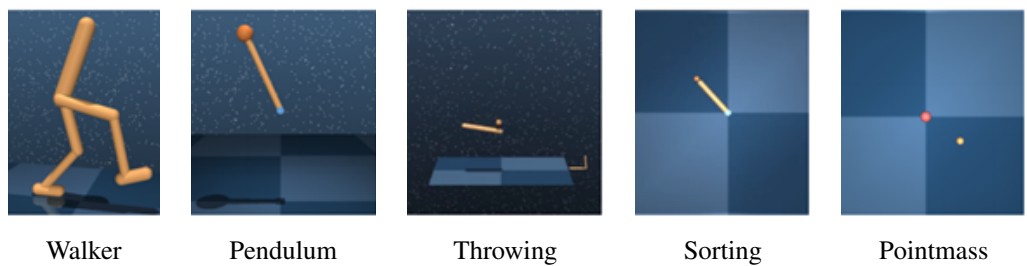

| Walker | Pendulum | Throwing | Sorting | Pointmass |

Figure 3: Five HIP-MDP tasks used in our experiments.

## 4.1 HIP-MDP TASKS

We evaluate our model on a variety of continuous control tasks from the DeepMind Control Suite (Tassa et al., 2018), along with some tasks developed in MuJoCo (Todorov et al., 2012). All tasks involve operating in a continuous control environment with varying physics. The tasks are as follows:

- DMC Walker Run - Make the Walker run as fast as possible in 2D, where the contact friction is variable.
- DMC Pendulum Swingup - Swing a pendulum to an upright position, where the pendulum mass is variable.
- Throwing - Control a paddle to throw a ball into a goal, where the ball mass is variable.
- Sorting - Move an object to a desired location, where the object mass is variable and the target location depends on the mass: heavier objects to the left and lighter objects to the right.
- DMC Pointmass - Move the point mass to the target location, where the x and y motors are randomly scaled. The target location depends on the motor scaling: away from the center for high motor scaling and towards the center for lower motor scaling.

When we design these tasks, we start by simply introducing randomization to the existing two tasks, DMC Walker Run and DMC Pendulum Swingup. Then, we purposely design the last two tasks, Sorting and DMC Pointmass, to incorporate a reward function that depends on their hidden parameters. Throwing also implicitly necessitates a policy for identifying the ball's mass and adjusting its trajectory. However, its reward function is not explicitly parameterized.

All the environments are visualized in Figure 3 and their randomization ranges are summarized in Table 1. A full description of all the environments used is in Section A in the appendix.

## 4.2 BASELINE ALGORITHMS

The baseline algorithms that we compare against are as follows:

- DreamerV2 : original DreamerV2 model proposed by Hafner et al. (2020).

| Method | Walker | Pendulum | Throwing | Sorting | Pointmass | Mean |
|---|---|---|---|---|---|---|
| PrivilegedDreamer | **766.20 ± 20.19** | **563.14 ± 147.44** | 788.59 ± 45.66 | **554.65 ± 26.25** | **670.23 ± 13.93** | **668.56 ± 70.87** |
| Dreamer + Decoder + ConditionedNet | 576.89 ± 96.68 | 329.80 ± 37.10 | 785.78 ± 64.18 | 180.85 ± 46.55 | 492.77 ± 17.82 | 473.22 ± 58.87 |
| Dreamer + Decoder | 671.85 ± 10.46 | 259.84 ± 26.08 | 707.51 ± 20.63 | 87.74 ± 43.24 | 480.96 ± 29.91 | 441.58 ± 28.21 |
| DreamerV2 (Hafner et al., 2020) | 715.57 ± 39.95 | 289.43 ± 214.12 | 706.09 ± 26.24 | 167.61 ± 33.38 | 488.41 ± 3.60 | 473.42 ± 99.26 |
| SAC (Haarnoja et al., 2018) | 475.22 ± 13.02 | 454.67 ± 268.98 | **945.65 ± 17.02** | 74.85 ± 88.03 | 393.49 ± 210.47 | 468.78 ± 158.03 |
| PPO (Schulman et al., 2017) | 79.73 ± 10.95 | 470.04 ± 324.05 | 707.03 ± 115.63 | 229.93 ± 181.12 | 545.86 ± 72.22 | 406.52 ± 176.93 |
| RMA (Kumar et al., 2021) | 75.28 ± 11.31 | 516.83 ± 386.43 | 624.57 ± 118.70 | 82.33 ± 416.57 | 545.31 ± 357.86 | 368.86 ± 305.00 |

Table 2: Model performance after 2 million timesteps of training

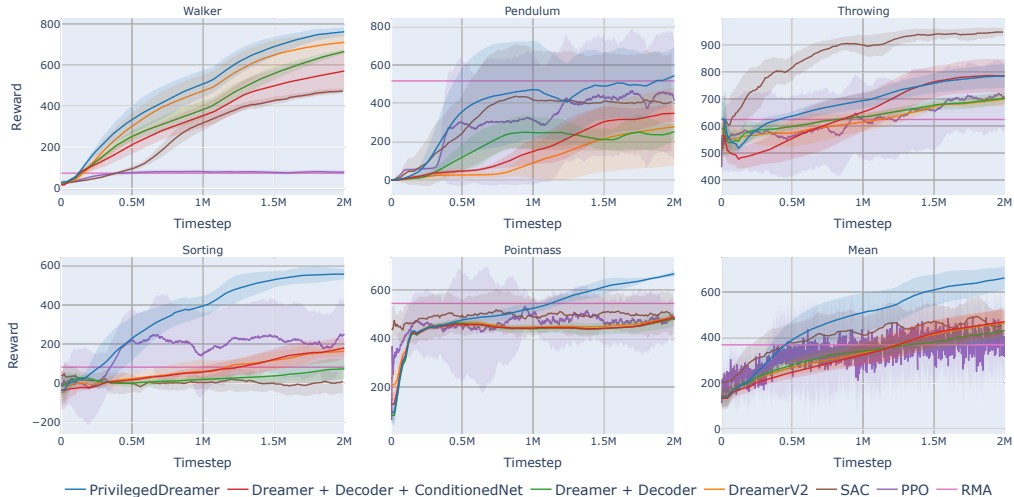

Figure 4: Learning curves for all tasks. PrivilegedDreamer shows the best performance against all the baseline algorithms, except for the throwing task that requires a very long horizon prediction.

- Proximal Policy Optimization (PPO): model-free, on-policy learning algorithm proposed by Schulman et al. (2017) using the implementation from Raffin et al. (2021).

- Soft Actor Critic (SAC): model-free, off-policy learning algorithm proposed by Haarnoja et al. (2018) using the implementation from Yarats & Kostrikov (2020).

- Rapid Motor Adaptation (RMA): model-free domain adaptation algorithm proposed by Kumar et al. (2021), which estimates hidden parameters from a history of states and actions. We train an expert PPO policy with $\omega$ as input and compare to the student RMA policy, which is trained with supervised learning to match $\omega$ using a history of previous states.

We select our baseline to cover all the state-of-the-art in model-based/model-free, on-policy/off-policy, domain randomization/adaptation algorithms. All models were trained for 2 million timesteps in each environment randomized as specified in Table 1.

To validate our design choices, we further evaluate the following intermediate versions of the algorithm.

- Dreamer + Decoder: This version only trains a decoder $\hat{\omega}_t \sim p_\phi(\hat{\omega}_t|h_t, z_t)$ by minimizing the hidden variable loss without an estimation module $\eta$. Also, $\hat{\omega}_t$ is not provided to the actor and critic and $h_t$ and $z_t$ are expected to contain all the information about the hidden parameter $\omega_t$.

- Dreamer + Decoder + ConditionedNet: This version is similar to the previous Dream + Decoder, but the estimated $\hat{\omega}_t$ is given to the actor and critic networks.

Note that the proposed PrivilegedDreamer can be viewed as the combination of Dreamer, an external estimation module, and conditioned networks trained with the hidden variable loss (PrivilegedDreamer = Dreamer + ExternalEstimation + Decoder + ConditionedNet).

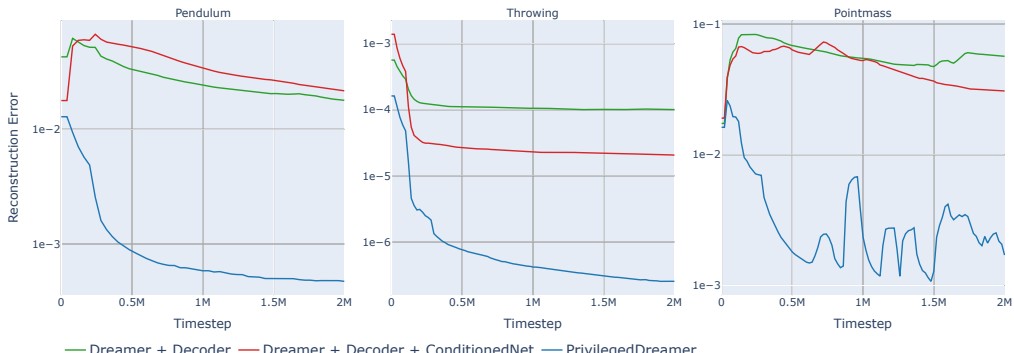

Figure 5: Hidden parameter reconstruction error during learning.

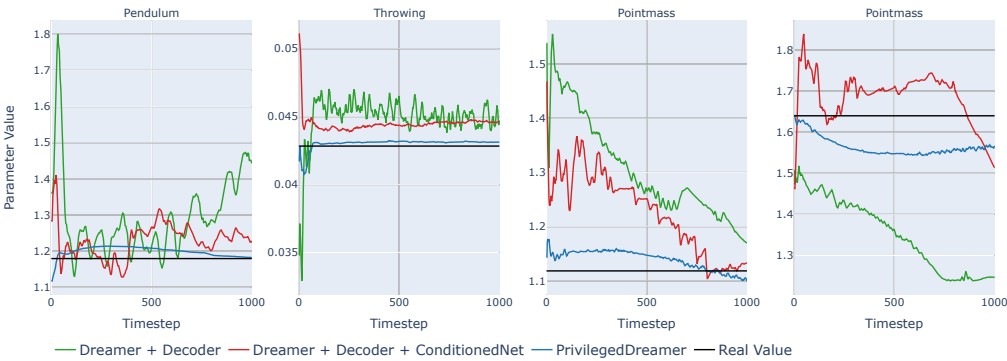

Figure 6: Online parameter estimation within an episode. The two estimated values for the Point-mass model are shown in separate plots to improve readability.

## 4.3 EVALUATION

**Performance**    To evaluate the effectiveness of the proposed method, we first compare the learning curves and the final performance of all the learned models. Learning curves for all models are shown in Figure 4, where the means and standard deviations are computed over three random seeds. Since RMA is trained in a supervised fashion using an expert policy and is not trained using on-policy environment interactions, we do not have a comparable learning curve, so we display the average performance as a horizontal line for comparison. Table 2 shows the average reward over 100 runs for each seed. We also report the average performance over five tasks in both Figure 4 and Table 2.

Overall, the proposed PrivilegedDreamer achieves the best average reward over five tasks. It shows a significant performance improvement over the second best model, vanilla DreamerV2, in both standard DM Control Suite tasks (Walker, Pointmass, Pendulum) as well as tasks we created ourselves (Sorting, Throwing). Performance margins are generally larger in the Sorting and DMC Pointmass tasks, where PrivilegedDreamer is the **only model** tested that does appreciably better than random. This is likely because the reward for these tasks explicitly depends on $\omega$ and DreamerV2 only implicitly adapts its behaviors to the hidden parameters. This indicates that a novel architecture of PrivilegedDreamer is effective for solving HIP-MDPs, particularly when the reward function is parameterized. We suspect that RMA and PPO do especially poorly on the Walker task because the 2 million timestep training limit is insufficient for on-policy algorithms. Similarly, we suspect that the small training size affects the ability of RMA to effectively adapt, and that it would be more competitive with our method with a larger training dataset, which our method does not need due to its better sample efficiency.

One notable outlier is the great performance of SAC on the Throwing task. We suspect that the nature of the problem makes it difficult for model-based RL algorithms, both PrivilegedDreamer and DreamerV2. In this task, a policy only has a few steps to estimate its hidden parameters and predict the ball's trajectory, which can easily accumulate model errors over a long time horizon. On the other hand, SAC, a model-free RL algorithm, efficiently modifies its behaviors in a model-free

fashion without estimating a ball trajectory. On-policy algorithms, PPO and RMA, are not sample-efficient enough to achieve good performance within two million steps.

**Hidden Parameter Estimation**    PrivilegedDreamer is based on the assumption that estimating hidden parameters is crucial for solving HIP-MDPs. Figure 5 illustrates the reconstruction errors during the learning process for the Pendulum, Throwing, and Pointmass tasks. In all cases, our PrivilegedDreamer exhibits faster convergence, typically within less than $0.5$ million environmental steps, resulting in more consistent learning curves. Additionally, Figure 6 displays the real-time estimation of hidden parameters during episodes. Our model accurately predicts these parameters within just a few steps, enhancing the performance of the final policies. These findings justify the effectiveness of an external LSTM-based hidden parameter estimation module.

### 4.4    Ablation Studies

Comparing our full PrivilegedDreamer model to the ablations, we see that our model is superior and each component is necessary for optimal performance. From Figure 5, we see that our full model is significantly better at reconstructing the hidden variable $\omega$ than Dreamer + Decoder + ConditionedNet, which is already better than Dreamer + Decoder. With this low reconstruction error, online estimation of $\omega$ is very effective, as shown in Figure 6, which shows that our method rapidly converges within 5% of the real value, while the ablated versions take longer to converge to a lower quality estimate. Specifically, our agents find near-correct hidden parameters at the beginning of the episodes within a few environmental steps in all scenarios, while the other baselines take more than 500 steps (Dreamer+Decoder+ConditionedNet in Pointmass) or converge to wrong values (Dreamer+Decoder in Pendulum and Pointmass).Using this high quality estimate of $\omega$ within our ConditionedNet, Figure 4 and Table 2 demonstrate that our method greatly outperforms the ablations. This validates our hypothesis that incorporating a good estimate of $\omega$ into the world model and policy networks improves the performance of a RL policy operating in an environment with variable $\omega$.

## 5    Conclusion

This paper presents a novel architecture for solving problems where the dynamics are dictated by hidden parameters. We model these problems with the Hidden parameter Markov Decision Process (HIP-MDP) and solve them using model-based reinforcement learning. We introduce a new model PrivilegedDreamer, based on the DreamerV2 world model, that handles the HIP-MDP problem via explicit prediction of these hidden variables. Our key invention consists of an external recurrent module to estimate these hidden variables to provide them as inputs to the world model itself. We evaluate our model on five HIP-MDP tasks, including both DeepMind Control Suite tasks and tasks we manually created where the reward explicitly depends on the hidden parameter, and found our model significantly outperforms the DreamerV2 model, as well as the other baselines we tested against.

Our research opens up several intriguing agendas for future investigation. Firstly, this paper has concentrated our efforts on studying hidden parameter estimation within proprioceptive control problems, intentionally deferring the exploration of visual control problems like Atari games or vision-based robot control for future works. We believe that the same principle of explicitly modeling hidden parameters can be effectively applied to these visual control challenges with minor adjustments to the neural network architectures. Furthermore, we plan to investigate more complex robotic control problems, such as legged locomotion (Wu et al., 2022), where real-world dynamics may be too sensitive to be precisely replicated by any of the hidden parameters. In such cases, we anticipate the need to devise better approximation methods. Lastly, we plan to delve into multi-agent scenarios in which these hidden parameters have an impact on the AI behavior of other agents. These subsequent research directions promise to extend the scope and impact of the original paper.

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

## A  ENVIRONMENT CONFIGURATION

All of the environments we use are from the DeepMind Control Suite Tassa et al. (2018) or were developed by us using MuJoCo Todorov et al. (2012). The full configuration of each environment is shown below. For each entry below, the dimension is shown in parentheses.

- DMC Walker Run

    - Observation space: Position (9), Velocity (9)
    - Action space: Joint angles (6)
    - Hidden parameter $\omega$: Contact friction (1)
    - Reward function: $r = r_{stand} * r_{move}$, where $r_{stand}$ rewards standing upright and $r_{move}$ rewards moving forward at a particular velocity

- DMC Pendulum Swingup

    - Observation space: Position (1), Velocity (1)
    - Action space: Pendulum angle (1)
    - Hidden parameter $\omega$: Pendulum mass scale (1)
    - Reward function: $r = r_{upright}$, where $r_{upright}$ rewards the pole being in an upright configuration

- Throwing

    - Observation space: Position (2), Velocity (2)
    - Action space: Throwing arm angle (1)
    - Hidden parameter $\omega$: Ball mass scale (1)
    - Reward function: $r = \exp\left(-|x - x_{goal}|\right)$, which gives a reward based on the distance to the goal position

- Sorting

    - Observation space: Position (1), Velocity (1)
    - Action space: Sorting arm angle (1)
    - Hidden parameter $\omega$: Sorting arm mass scale (1)
    - Reward function: $r = \exp\left(-|x - x_{goal}|\right)$, which gives a reward based on the distance to the goal position
    - The goal position $x_{goal}$ is determined as follows: $\begin{cases} +0.2 & \text{if } \omega < 0.6, \\ -0.2 & \text{otherwise.} \end{cases}$

- DMC Pointmass Easy

    - Observation space: Position (2), Velocity (2)
    - Action space: Pointmass motors (2)
    - Hidden parameter $\omega$: Pointmass motor scales (2)
    - Reward function: The reward function depends on $\omega$ and is defined as follows: $r = \begin{cases} \exp\left(-|x - x_{\text{goal}}|\right) & \text{if } \sum \omega < 3, \\ \left(1 - \exp\left(-|x - x_{\text{goal}}|\right)\right) & \text{otherwise.} \end{cases}$

## B DREAMER HYPERPARAMETERS

| Parameter | Value |
|---|---|
| Replay buffer size | $5 \times 10^5$ |
| $\gamma$ discount | 0.99 |
| Batch size | 1024 |
| Learning rate (encoder, decoder, and LSTM) | $10^{-4}$ |
| Learning rate (actor) | $2 \times 10^{-4}$ |
| Learning rate (critic) | $8 \times 10^{-5}$ |
| Hidden layers (all networks) | [400, 400, 400, 400] |
| Hidden dimension (actor and critic) | 1024 |
| Discount $\lambda$ | 0.95 |
| Gradient clipping threshold | 100 |
| Imagination horizon | 15 |
| LSTM hidden dimension | 64 |

## C PPO HYPERPARAMETERS

| Parameter | Value |
|---|---|
| Learning rate | $3 \times 10^{-4}$ |
| Steps per update | 2048 |
| Batch size | 64 |
| $\gamma$ discount | 0.99 |
| GAE $\lambda$ | 0.95 |
| Clip range | 0.2 |
| Gradient clipping threshold | 0.5 |

## D SAC HYPERPARAMETERS

| Parameter | Value |
|---|---|
| Replay buffer size | $10^6$ |
| Seed steps | 5000 |
| Initial temperature | 0.1 |
| $\gamma$ discount | 0.99 |
| Batch size | 1024 |
| Learning rate (all networks) | $10^{-4}$ |
| Hidden layers (actor and critic) | [1024, 1024] |
| Critic $\tau$ | 0.005 |

# E  RMA HYPERPARAMETERS

| Parameter | Value |
|---|---|
| Encoder shape | [256, 128] |
| History size | 25 |
| Train size | 100000 |
| Learning rate | 0.004 |
| Activation | ELU Clevert et al. (2015) |

