# OpenReview forum: "PrivilegedDreamer: Explicit Imagination of Privileged Information for Adaptation in Uncertain Environments"
_ICLR.cc/2024/Conference — Submitted to ICLR 2024_

### Official Review · Reviewer_TuCu · 2023-10-24

**Soundness:** 3 good
**Presentation:** 3 good
**Contribution:** 2 fair
**Rating:** 5
**Confidence:** 5

**Summary:**

The paper proposes a version of RMA (Kumar'21) applied to a model-based algorithm (Dreamer). There is a network that regresses privileged MDP parameters (e.g. object friction). The dynamics model, actor, and the critic are conditioned on the regressed parameters. This allows to more easily solve tasks where privileged parameters are known during training but not test time. A simple benchmark is proposed based on mujoco tasks and the method outperforms existing baselines in 2 of the 5 tasks.

**Strengths:**

- The method is intuitive and relevant

**Weaknesses:**

- The method is incremental
- The performance is mediocre. The method outperforms baselines only in 2 tasks out of 5
- The benchmark doesn't seem fit for evaluating methods that leverage hidden parameters. RMA, which SOTA for hidden parameter estimation, only outperforms the other baselines in 1 of the 5 tasks. In other tasks, PPO, SAC, or Dreamer work better than RMA. This suggests estimating hidden parameters is not important for these tasks.
- Alternatively, it is possible better baselines are needed.
- There seems to be little discussion of how this method is better than RMA, which is a strange omission since that's the most important baseline.
- The method is only evaluated on a toy benchmark. Since RMA is the main baseline, it would be appropriate to compare to existing benchmarks RMA was evaluated on.

**Questions:**

See weaknesses

---

### Official Review · Reviewer_TWC6 · 2023-10-26

**Soundness:** 2 fair
**Presentation:** 3 good
**Contribution:** 2 fair
**Rating:** 3
**Confidence:** 4

**Summary:**

The work introduces, what the authors call, PRIVILEGED DREAMER. These deep ANN architecture augments the DREAMER architecture by Hafner and colleagues with a recurrent neural network module that is designed to extract hidden parameters from video data. In particular, the hidden parameters refer to not directly observable properties of objects in a scene, such as their mass. These parameters are naturally inferable from observing their interaction dynamics with other objects. This fact is exploited by the presented architecture. Moreover, the work is theoretically accessibly grounded in a previous formalization of "Hidden-parameter MDP", which is essentially a characterization of a particular type of POMDP, which turns into an MDP when hidden parameters are known or have been inferred. The paper considers three dynamic tasks from the DeepMind Control Suite and two designed once via the MuJoCo simulator. Latent parameters that are varied are friction, mass scaling, and motor scaling factors.

**Strengths:**

The paper adds a simple LSTM module and improves performance in 3-4 of the 5 considered tasks.
The addition is appealingly grounded in a theoretical framework (Hidden-parameter MDP).
The paper is accessibly written.
All information for reproducibility is provided.
Ablation studies are included.

**Weaknesses:**

The improvements are not overly strong.

In the conclusion the authors state that their model does outperform all other baselines, which is not true for the throwing task.

Most important weakness: the values of the hidden parameters are provided in the loss function (if I am not fully mistaken – this apparent fact is not made very explicit – but the log likelihoods in the loss function appear to assume knowledge about the true values). As a result, dreamer needs to be beaten. In fact, I consider the comparison with Dreamer somewhat inappropriate, because you train your model to learn about key information. [Note that even if you do not provide this information somehow – but then how is Fig. 6 created (?) – I am not fully convinced of the value of the approach due to the other weaknesses.]

Another valuable comparison – as an upper bound – would be a hidden-parameter informed DREAMER.

I am surprised that the authors did not analyze (or optimize further) how the omega is fed into the subsequent modules. In particular, and as a (hopefully useful) suggestion, I would recommend introducing one embedding layer between the omega output and its input to the forward model / the policy+value function modules, because the physical value is probably not ideally suited for those respective models.

The tasks are relatively simple. The provided input appears to be “Proprioceptive” – as the authors write in the conclusion – so giving rather precise state information of the controlled walker / pendulum / etc… it remains fully unclear how this system scales to more challenging environments with many hidden variables – possibly also where some of these variables are not even relevant.

It was not analyzed, which inputs are best suited to be streamed into the new hidden parameter estimation module. As another (hopefully useful) suggestion, I can imagine that the error signal, that is the difference between the outputs and the subsequent inputs, would be very informative.

**Questions:**

What is the exact input and target output (x_t ) for your architecture in the respective problems? I think this should be clarified in the main text.

What is the motivation for estimating omega twice (head vs. tilde).

Reconstruction error in figure 5 is computed how (averaged over what, MSE?)? Shouldn’t this be the log probability?

Figure 6 – could it distinguish between omega-tilde and omega-head?

In the conclusions, wouldn’t it be the main challenge to learn hidden parameters from scratch without providing their values during training – but under the assumption that their values are stable over the complete episode. This can be done, for example, via retrospective inference mechanisms.

---

### Official Review · Reviewer_vccC · 2023-10-29

**Soundness:** 2 fair
**Presentation:** 3 good
**Contribution:** 2 fair
**Rating:** 3
**Confidence:** 4

**Summary:**

This paper proposes a solution approach to a family of problems called hidden parameter MDPs (HIP-MDPs) where a set of unknown (hidden) parameters influence the transition function and reward function. The proposed approach, called PrivilegedDreamer, builds upon DreamerV2 model-based RL approach by explicitly estimating hidden parameters in the environments using recent history of action-observation pairs. They argue that such explicit representation and learning of hidden parameters is essential for synthesizing effective, sample-efficient policies in problems that can be modeled as HIP-MDPs. The authors evaluate their model on 5 continuous control tasks to support their claim that the proposed approach outperforms the original Dreamer model and several other baselines.

**Strengths:**

- The proposed approach is conceptually simple and can be easily combined with many model-based RL approaches beyond Dreamer-like architectures.

- For domains/environments with stationary hidden parameters, the hidden parameter estimates can be effectively reused for other policy synthesis on other tasks with different objectives on the same environment.

**Weaknesses:**

- The experimental results from a small set of tasks does not convincingly demonstrate the approach's superiority over existing methods. For example, past approaches like DreamerV2 provide data from extensive experiments on a variety of environments. While I acknowledge computational constraints, I believe that limiting training to 2 million steps (as currently done in the paper) should make it feasible to test on more environments. Ideally, I think the authors should release the code for reproducing the results or at the very least for the simulation environments the authors tested on as they include two non-DMC custom environments.


- Within the experiments performed, I have concerns about the statistical significance of the results presented and some of the inconsistencies in the text/figures:
    - For throwing environment, the range of the hidden parameter is mentioned as [0.2 - 1.0] in Table 1 yet in Figure 6, which shows an instance of online parameter estimation within an episode, the real-value of the same parameter seems to be 0.042 which is out of this range.
    - Authors mention that SAC has the best performance in throwing task as the policy doesn’t have a lot of steps to estimate its hidden parameters but Figure 6 seems to show that the hidden parameter estimate is fairly accurate in less than 100 steps. In fact, part of the text mentions “our agent finds near-correct hidden parameters … with a few environmental steps in all scenarios …”
    - If I understand it correctly, RMA can also provide estimate of the hidden parameter with an appropriate choice of extrinsic vector encoding dimensionality but it seems to be missing from figure 6.
    - Using only three random seeds might be a bit low.
- The paper provides insufficient details about the implementation of baseline approaches. This lack of information is critical, given that the authors introduce new tasks for evaluating their proposed approach, and baseline results for these tasks aren't available in existing literature. Consequently, the significance of the proposed approach's performance hinges on the extent to which the baseline approaches were reasonably tuned. Additionally, Appendix E's hyperparameter details appear incomplete, particularly for RMA, with some key parameters unmentioned. The justification for selecting specific values, like the seemingly arbitrary learning rate of 0.004, is also absent. This issue is particularly important given that the performance margins between the proposed approach and the baselines are narrow for some of the five tasks tested.
- The related work section is very terse, lacking an in-depth discussion on how this work is similar to or different from prior approaches including the approaches that learn a context vector to capture extrinsics/environment information from history.
- I am not entirely convinced by the motivation of problems with hidden parameters influencing their reward function. It would strengthen the paper if the authors could better motivate their problem formulation, perhaps by providing examples of real-world scenarios that fit this model.
- I have some reservations about the novelty of the paper, particularly since the concept of explicit hidden parameter estimation, exemplified by methods like RMA, is already established in the field. The combination of this idea with DreamViewer might seem incremental. However, my overall assessment isn't affected by this aspect.

**Questions:**

In addition to the questions raised in the weaknesses section, I have the following questions:

- Can the authors clarify how much effort was put into tuning the baseline approaches and what was the reasoning behind some of the design choices in their architecture parameters?
- Is there a particular reason why 2 million steps was selected as cut-off for evaluation? Or was it arbitrary?
- Does the average performance shown in Figure 5 of RMA and the values reported in table 2 correspond to RMA trained with 2 million steps worth of data? Was the adaptation module/extrinsics estimator trained using the same data?
- How would the performance be when have an explicit hidden parameter estimation module that feeds into RSSM but no Decoder or ConditionedNet?

---

### Official Review · Reviewer_iDRt · 2023-10-31

**Soundness:** 2 fair
**Presentation:** 2 fair
**Contribution:** 1 poor
**Rating:** 3
**Confidence:** 5

**Summary:**

The authors propose an RL method for solving Hidden-Parameter MDP tasks, where dynamics and rewards depend on some unobserved parameter (e.g. friction). One strategy is to estimate the hidden parameters from history, and then adapt and behave accordingly to the estimates. The authors bring this intuition into MBRL by extending DreamerV2 to estimate the hidden parameters and conditioning various components on it (policy, critic etc.). This allows their method, PrivilegedDreamer, to better solve the HiMDP tasks.

**Strengths:**

Method is quite simple. The overall idea of estimating the hidden parameters and conditioning the various  agent networks on the estimates makes sense.

An interesting experimental result is that for one task, model-free RL which does not estimate parameters does the best. This shows that for some HiMDPs, explicit estimation of hidden parameters is too hard, so learning a robust policy is better.

**Weaknesses:**

##  RL tasks seem too toy and easy.
The tasks are toy. They are 2D,  low dimensional states, from Deepmind Control. They do not show any harder tasks on the DMC, like Humanoid, etc. Note that related work, like CaDM [1] do show harder DMC environments (humanoid, ant, higher dimensional state / action spaces).  The Physics randomization is only done on one variable for each task. Why not have tasks with multiple variables for randomization? This seems artificial, in reality, many system parameters need to be identified simultaneously.  In summary, given the limited evaluation on easy tasks, there is little evidence to believe that this will scale to actually hard tasks.

## Weak / Ill-fitting baselines
Their choice of baselines is questionable, authors seemed to have chosen deliberately weak baselines or baselines not suited for their problem setting. For many of their baselines they choose, there is a concern that is obvious to me.

First, their experiments are in low-sample regimes. Many of their baselines, which were not designed for such regimes, will fail. While this is a valid point to make, it is quite obvious that PPO will  not work with only 2M steps, for example. SAC is an okay RL baseline, but some SOTA sample-efficient model-free RL baseline like RedQ / DroQ would be the best.

The authors also compare against DreamerV2, a sample-efficient MBRL agent. Once again, it is okay, but a comparison against DreamerV3 would be more competitive and convincing if PrivilegedDreamer beats it.

__Weak Domain Adaptation Baseline__ This brings up the most important point - the most important baseline is a domain adaptation baseline, which will show how PrivilegedDreamer compares against a method that is also doing hidden parameter estimation. However, they choose RMA for the domain adaptation baseline, which uses sample-inefficient PPO for  sim2real robotics applications. RMA is not meant to be evaluated in a sample-efficient setting, since it hinges on using fast simulators and lots of samples to train privileged policies.

The authors should be fair to RMA, and allow it to run for many more steps to allow its privileged PPO policy to converge. While this baseline is unfair in the amount of samples, it is a useful upper-bound baseline. It would be good if PrivilegedDreamer can be close, match, or outperform this version of RMA.

There do exist several sample-efficient domain adaptation baselines, that similar to PrivilegedDreamer, estimate hidden parameters in a model-based RL framework. The authors mention Context Aware Dynamics Models [1], which is evaluated on harder tasks than PrivilegedDreamer and has a public codebase. I can also suggest VariBAD [2], another model-based domain adaptation algorithm with a public codebase. In short, the authors should choose a fair domain adaptation baseline that is designed for the low-sample regime, and has similar design / structure to Privileged Dreamer.

## Limited Novelty
The main novelty seems to be taking the well-known strategy for domain adaptation (identifying system parameters and conditioning the policy on it) and implementing it into Dreamer. So there doesn't seem to be much actual new innovation in the conceptual space of methods that tackle HiPMDPs.

[1] Lee, Kimin, et al. "Context-aware dynamics model for generalization in model-based reinforcement learning." International Conference on Machine Learning. PMLR, 2020.

[2] Zintgraf, Luisa, et al. "Varibad: A very good method for bayes-adaptive deep rl via meta-learning." arXiv preprint arXiv:1910.08348 (2019).

**Questions:**

Can the authors make the experimental section more convincing? For example, [1] proposes several more high dimensional tasks (SlimHumanoid, Ant, etc.). Next, can the authors choose more competitive baselines, especially for baselines that do domain adaptation?

It would be helpful if the authors can compare and contrast PrivilegedDreamer against other similar model-based domain adaptation baselines.

---

### Meta-Review · Area_Chair_Ctz3 · 2023-12-06

**Metareview:**

Summary: This paper proposes a model-based RL method for solving Hidden-Parameter MDPs wherein the dynamics and rewards depend on some unobserved parameter (e.g. friction). The key idea is to estimate the hidden parameter from a history of observations. The proposed approach is compared to one other approach which performs similar estimation (e.g., RMA) and Dreamer and PPO / SAC baselines in simulated HiMDP MuJoCo examples.

Strengths: The idea is very simple and intuitive, there are several ablation studies, and the problem is clearly formalized within the well-studied Mi-MDP modeling framework.

Weaknesses: All reviewers share the same concern that the core technical idea is somewhat incremental (considering that there are many other works which leverage estimators for unknown parameters, like friction), the tasks are relatively simple and many non-adaptive baselines perform well too (raising the question of if the hidden parameter estimation is necessary in these tasks), and the baselines that perform adaptation could be much stronger.

**Justification For Why Not Higher Score:**

After I carefully reviewed the manuscript and the reviewer’s comments, I agree with the reviewer's reasons for recommending rejection.

**Justification For Why Not Lower Score:**

N/A

---

### Decision · Program_Chairs · 2024-01-16

Reject